# Quantum Mpemba Effect from Non-Normal Dynamics

**DOI:** 10.3390/e27060581

**Published:** 2025-05-29

**Authors:** Stefano Longhi

**Affiliations:** 1Dipartimento di Fisica, Politecnico di Milano, Piazza L. da Vinci 32, I-20133 Milano, Italy; stefano.longhi@polimi.it; 2IFISC (UIB-CSIC), Instituto de Fisica Interdisciplinar y Sistemas Complejos, E-07122 Palma de Mallorca, Spain

**Keywords:** Mpemba effect, open quantum systems, waveguide quantum electrodynamics

## Abstract

The quantum Mpemba effect refers to the counterintuitive phenomenon in which a system initially farther from equilibrium relaxes faster than one prepared closer to it. Several mechanisms have been identified in open quantum systems to explain this behavior, including the strong Mpemba effect, non-Markovian memory, and initial system–reservoir entanglement. Here, we unveil a distinct mechanism rooted in the non-normal nature of the Liouvillian superoperator in Markovian dynamics. When the Liouvillian’s eigenmodes are non-orthogonal, transient interference between decaying modes can induce anomalous early-time behavior—such as delayed thermalization or transient freezing—even though asymptotic decay rates remain unchanged. This differs fundamentally from strong Mpemba effects, which hinge on suppressed overlap with slow-decaying modes. We demonstrate this mechanism using a waveguide quantum electrodynamics model, where quantum emitters interact with the photonic modes of a one-dimensional waveguide. The directional and radiative nature of these couplings naturally introduces non-normality into the system’s dynamics. As a result, certain initial states—despite being closer to equilibrium—can exhibit slower relaxation at short times. This work reveals a previously unexplored and universal source of Mpemba-like behavior in memoryless quantum systems, expanding the theoretical framework for anomalous relaxation and opening new avenues for control in engineered quantum platforms.

## 1. Introduction

The thermal relaxation of physical systems is commonly assumed to follow an intuitive rule: states closer to equilibrium should relax faster than those further away. Rooted in classical thermodynamics, this logic suggests that, for instance, a hotter object should cool more slowly than a colder one when coupled to a thermal bath. However, this expectation can break down. One striking example is the Mpemba effect (ME), where a system initially at a higher temperature cools faster than a similar system at a lower temperature [1,2,3,4,5,6,7,8,9,10,11,12]. While its classical version remains debated and lacks a universally accepted explanation [4,12], the Mpemba effect has sparked renewed interest in the context of quantum systems, where it can be explored in cleaner, more controllable settings (for a recent review, see ref. [13]). This renewed focus has led to a wide range of theoretical and experimental investigations in both open [14,15,16,17,18,19,20,21,22,23,24,25,26,27,28,29,30,31,32,33,34,35,36] and closed [37,38,39,40,41,42,43,44,45,46] quantum systems, spanning diverse platforms such as spin systems, dissipative quantum circuits, and open quantum many-body dynamics. Broadly speaking, the Mpemba effect refers to scenarios in which a state initially farther from equilibrium relaxes faster than one closer to it. Several mechanisms underlying this behavior have been identified. The strong ME arises when certain initial states have vanishing overlap with the slowest-decaying eigenmodes of the Liouvillian, leading to enhanced asymptotic decay [13,15,17,18,19,21,29]. Another pathway involves initial system–environment correlations, such as entanglement, which effectively alter the open system’s initial conditions, even under Markovian assumptions [34]. Non-Markovian dynamics offer yet another route, where memory effects can produce anomalous relaxation [9,36], including cases where a state reaches equilibrium within the environment’s memory time—a phenomenon referred to as the extreme quantum Mpemba effect [36]. Beyond open quantum systems, Mpemba-like phenomena have also been observed in closed quantum systems evolving unitarily after a quantum quench [28,37,38,39,43,44]. In such cases, anomalous relaxation behavior has been identified in the growth of entanglement entropy, where a system starting further from equilibrium can approach a thermal-like state faster than one closer to equilibrium. These mechanisms illustrate that the ME in quantum systems can emerge from a variety of physical mechanisms.

In this work, we introduce a new and conceptually distinct mechanism that can give rise to a Mpemba-like effect in fully Markovian quantum dynamics, without relying on long-time decay rate hierarchies, non-Markovianity, or entanglement. The key ingredient is the non-normality of the Liouvillian superoperator governing the system’s evolution. Non-normal operators—those that do not commute with their adjoint—have non-orthogonal eigenmodes, leading to transient interference effects between decaying components [47,48,49]. As a consequence, certain initial conditions can exhibit anomalous early-time behavior, including slow early relaxation, transient freezing dynamics, or delayed relaxation [48,49,50,51,52,53,54,55,56,57], even when the long-time asymptotic decay is exponential with the decay rate of the slowest decaying mode. We refer to this phenomenon as the non-normal quantum ME. To illustrate this effect, we study a model system drawn from waveguide quantum electrodynamics (QEDs), in which quantum emitters (modeled as two-level systems) couple to the photonic modes of a one-dimensional waveguide [58,59,60,61]. Such setups facilitate long-range coherent and incoherent interactions mediated by long-lived propagating photons and naturally give rise to non-normality in the Liouvillian. We analyze the relaxation dynamics of different initial states and identify regimes where certain configurations are closer to equilibrium, evolving under memoryless, Lindblad-type dynamics. These exhibit slower short-time relaxation due to destructive modal interference, leading to a pronounced ME. Our results enrich the taxonomy of quantum ME by identifying transient interference—rather than asymptotic behavior—as a key source of anomalous relaxation. This opens new directions for controlling nonequilibrium processes in engineered quantum systems, particularly in photonic or circuit QED platforms where non-normality can be designed and exploited for quantum state control or thermodynamic manipulation.

## 2. Mpemba Effect from Non-Normal Dynamics in Markovian Open Quantum Systems

Let us consider a Markovian open quantum system whose state at time *t* is described by the reduced density operator ρ(t). Its time evolution is governed by the Liouvillian superoperator L in the standard Lindblad form [62], (dρ/dt)=Lρ(t), where(1)$$L\rho =-i\left[H,\rho \right]+\sum_{k}\left(L_{k}\rho L_{k}^{†}-\frac{1}{2}\left\{L_{k}^{†}L_{k},\rho \right\}\right).$$
In the above equation, *H* is the system Hamiltonian describing the intrinsic unitary evolution of the system, whereas $L_{k}$ are a set of jump operators describing the dissipative part of the dynamics. The Liouvillian superoperator can be equivalently written as follows:(2)$$L\rho =-i\left(H_{eff}\rho -\rho H_{eff}^{†}\right)+\sum_{k}L_{k}\rho L_{k}^{†}\;,$$
where(3)$$H_{eff}=H-\frac{i}{2}\sum_{k}L_{k}^{†}L_{k}$$
is an effective non-Hermitian Hamiltonian, and the last term on the right hand side of Equation (Equation 2), $\sum_{k}L_{k}\rho L_{k}^{†}$, describes quantum jumps. We assume that the Liouvillian L is diagonalizable, and let us denote by $\lambda_{\alpha }$ the eigenvalues of L, and by $\rho_{\alpha }$ and $l_{\alpha }$ the corresponding right and left eigenvectors, respectively. Since the eigenvalues have a nonpositive real part, they can be sorted in descending order, 0=λ0>Re(λ1)≥Re(λ2)≥…. We assume that the zero eigenvalue $\lambda_{0}=0$ is not degenerate, indicating that the steady state is unique and corresponds to the state $\rho_{0}$. The Liouvillian gap is defined as Δ=−Re(λ1). The right and left eigenvectors of L form an orthogonal basis, namely $\langle l_{\alpha }|\rho_{\beta }\rangle =\delta_{\alpha ,\beta }$, where for two operators *A* and *B*, the scalar product 〈A|B〉 is defined as $\langle A|B\rangle =\mathrm{Tr}\left(A^{†}B\right)$ (see e.g., [53]). For an initial state $\rho_{i}$, the general evolution of the density operator ρ(t) can be written as follows:(4)$$\rho \left(t\right)=\exp\left(Lt\right)\rho_{i}=\rho_{0}+\sum_{\alpha \geq 1}c_{\alpha }\rho_{\alpha }\exp\left(\lambda_{\alpha }t\right)\;,$$
where $c_{\alpha }=\langle l_{\alpha }|\rho_{i}\rangle =\mathrm{Tr}\left(l_{\alpha }^{†}\rho_{i}\right)$. To quantify the distance d(t) of the system from its equilibrium state $\rho_{0}$ during the relaxation process, different measures can be assumed (see e.g., [13,63]), such as the trace distance $d\left(t\right)=d_{T}\left(t\right)=\left(1/2\right)\mathrm{Tr}\sqrt{A^{†}A}$ and the Frobenius (Hilbert-Schmidt) distance $d\left(t\right)=d_{F}\left(t\right)=\sqrt{\mathrm{Tr}(A^{†}A)}$, where $A=\rho \left(t\right)-\rho_{0}$. For example, assuming the Frobenius distance $d\left(t\right)=d_{F}\left(t\right)$, using Equation (Equation 4), one has the following:(5)$$d^{2}\left(t\right)=\sum_{\alpha ,\beta =1,2,3,\ldots }c_{\alpha }^{*}c_{\beta }\exp\left[\left(\lambda_{\alpha }^{*}+\lambda_{\beta }\right)t\right]\mathrm{Tr}\left(\rho_{\alpha }^{†}\rho_{\beta }\right).$$
The ME arises when comparing the relaxation dynamics of two initial states, I and II, described by the density operators $\rho_{i}^{\left(I\right)}$ and $\rho_{i}^{\left(II\right)}$. Assuming that state I is initially further away from equilibrium, i.e., $d^{\left(I\right)}\left(0\right)>d^{\left(II\right)}\left(0\right)$, the ME occurs whenever, after an initial transient, one has $d^{\left(I\right)}\left(t\right)<d^{\left(II\right)}\left(t\right)$ for $t>t_{f}$, $t_{f}$ being a suitably large time. Since the Liouvillian spectral gap Δ determines the decay rate of the slowest relaxation mode, it is naturally expected that its inverse, 1/Δ, sets the characteristic timescale for relaxation. This expectation holds when the Liouvillian superoperator L is *normal*, i.e., when it commutes with its adjoint: $[L,L^{†}]=0$. In this case, the right and left eigenvectors of L do coincide; thus, right eigenvectors corresponding to distinct eigenvalues are orthogonal, meaning $\langle \rho_{\alpha }|\rho_{\beta }\rangle =\mathrm{Tr}\left(\rho_{\alpha }^{†}\rho_{\beta }\right)=0$ for α≠β. As a result, the interference terms with α≠β on the right-hand side of Equation (Equation 5) vanish, yielding the following:(6)d2(t)=∑α,1,2,3,…|cα|2exp2Re(λα)t.
This result clearly demonstrates that the inverse of the spectral gap, 1/Δ=1/|Re(λ1)|, indeed sets an upper bound on the relaxation time toward equilibrium. More precisely, the actual relaxation time is determined by the largest value of 1/|Re(λα)| for which the corresponding spectral coefficient $c_{\alpha }$ is non-zero. When the initial state displays a non-vanishing spectral coefficient $c_{1}\neq 0$, the relaxation time is given by 1/Δ, i.e., it is limited by the spectral gap. When the coefficient $c_{1}$ vanishes, the relaxation proceeds faster, governed by the next slowest decay rate, −Re(λ2). This is the essence of the strong Mpemba effect (see, e.g., [13,15,18,21,29]): if state I, initially farther from equilibrium, does not excite the slowest decaying mode (i.e., $c_{1}=0$ for state I), it relaxes faster at long times than state II, which does have a non-zero overlap with this slowest mode (i.e., $c_{1}\neq 0$ for state II). This situation is schematically illustrated in Figure 1a.

However, when L is not a normal operator, i.e., $LL^{†}\neq L^{†}L$, the inverse of the spectral gap 1/Δ *does not set* an upper limit to the characteristic relaxation time toward equilibrium, which *can be much longer* than 1/Δ. Indeed, several studies have emphasized that the actual relaxation time is not necessarily bounded by the Liouvillian gap [53,54,55,56], although the latter does determine the asymptotic decay rate in the long-time limit [15]. This implies that, in the transient regime, the system can exhibit decay rates that are slower than the Liouvillian gap would suggest. Such anomalously slow relaxation, which arises from the interference terms on the right hand side of Equation (Equation 5), can give rise to Mpemba-like effects without requiring accelerated thermalization for one of the two initial states. This type of ME, schematically illustrated in Figure 1b, is fundamentally different from the strong ME and may be referred to as the *non-normal Mpemba effect*, as it originates from the non-normal nature of the Liouvillian superoperator.

## 3. Non-Normal Mpemba Effect in Waveguide Quantum Electrodynamics

To illustrate the non-normal Mpemba effect within a key quantum platform, we consider the relaxation dynamics of quantum emitters in the context of waveguide quantum electrodynamics (waveguide QED). In these systems, quantum emitters such as atoms, quantum dots, or superconducting qubits are coupled to one-dimensional photonic waveguides, enabling strong and tunable light–matter interactions [58,59,60,61]. Waveguide QED platforms are particularly well-suited for exploring open quantum system dynamics, as they naturally give rise to structured reservoirs, long-range photon-mediated interactions, and non-Markovian effects. These features naturally lead to a non-normal Liouvillian superoperator, where the relaxation dynamics are governed not only by the spectral gap but also by the non-orthogonality and interference of decay modes. This makes waveguide QED an ideal setting for observing the non-normal ME, where anomalously slow relaxation emerges even in the absence of accelerated decay pathways. Waveguide QED systems offer a broad range of physical implementations [60], including atoms coupled to photonic waveguides [64,65], photonic crystals [66], quantum dots embedded in nanophotonic structures [67], and superconducting circuits [68,69,70,71]. These platforms provide fertile ground for investigating collective phenomena and complex thermalization behavior driven by coherent and incoherent photon-mediated interactions over long distances.

We consider the dissipative dynamics of a set of *N* point-like quantum emitters, described as two-level (spin) systems, coupled to a one-dimensional bosonic waveguide, as schematically illustrated in Figure 1c. In the Born–Markov approximation, neglecting retardation effects, the evolution of the reduced density matrix ρ(t) of the quantum emitters is governed by the Lindblad master equation [58,59,62]:(7)$$\frac{d\rho }{dt}=L\rho =-i\left(H_{eff}\rho -\rho H_{eff}^{†}\right)+\gamma_{R}L_{R}\rho L_{R}^{†}+\gamma_{L}L_{L}\rho L_{L}^{†}\;,$$
where(8)$$\begin{array}{ccc} H_{eff} & = & \sum_{n=1}^{N}\left(\omega_{n}-\omega_{0}-i\frac{\gamma_{L}+\gamma_{R}}{2}\right)\sigma_{n}^{†}\sigma_{n}\\ & - & i\gamma_{L}\sum_{n<l}\sigma_{n}^{†}\sigma_{l}\exp\left[ik\left(x_{l}-x_{n}\right)\right]-i\gamma_{R}\sum_{n>l}\sigma_{n}^{†}\sigma_{l}\exp\left[ik\left(x_{n}-x_{l}\right)\right] \end{array}$$
is the effective non-Hermitian Hamiltonian, and(9)$$L_{L}=\sum_{n=1}^{N}\sigma_{n}\exp\left(ikx_{n}\right)\;,\;\;L_{R}=\sum_{n=1}^{N}\sigma_{n}\exp\left(-ikx_{n}\right)$$
are the jump operators associated to decay into the left- and right-propagating modes of the waveguide. In the above equations, $x_{n}$ is the spatial position of the *n*-th quantum emitter in the one-dimensional waveguide, with $x_{1}<x_{2}<\ldots <x_{N-1}<x_{N}$ (Figure 1c), $\omega_{n}$ is its resonance frequency, close to the frequency $\omega_{0}=kc$ of the excited left/right propagating photonic modes in the waveguide, $\sigma_{n}^{†}={|e\rangle }_{n}\langle g|$ and $\sigma_{n}={|g\rangle }_{n}\langle e|$ are the atom transition operators with ground state |g〉 and excited state |e〉, and $\gamma_{L,R}$ are the emission rates of a single quantum emitter into the right (R) and left (L) propagating modes of the waveguide. A chiral waveguide corresponds to $\gamma_{R}\neq \gamma_{L}$ [58]. The Hilbert space of the *N* quantum emitters is spanned by the $2^{N}$ states $|q_{1}\rangle ⊗|q_{2}\rangle ⊗\ldots |q_{N}\rangle$, where $|q_{n}\rangle$ is either |g〉 or |e〉, corresponding to the *n*-th atom being in the ground or excited state. On this basis, the density operator ρ(t) is described by a $2^{N}\times 2^{N}$ matrix, and the superoperator L is described by a $2^{2N}\times 2^{2N}$ matrix. We assume that there are not atom–photon bound states, so that any initial atomic excitations irreversibly decay into the ground states, corresponding to the unique stationary state $\rho_{0}=\left|0\rangle \langle 0\right|$, where |0〉≡|g〉⊗|g〉⊗…⊗|g〉 is the pure state with all *N* atoms in the ground state.

To unveil the emergence of the non-normal ME, let us restrict our attention considering the states corresponding to up to a single excitation in the system [60], so that the Hilbert space is greatly reduced and comprises solely the (N+1) states |0〉 (the ground equilibrium state) and $|\alpha \rangle \equiv \sigma_{\alpha }^{†}|0\rangle$ (the α-th atom is in the excited state, while all other atoms are in the ground state, with α=1,2,…,N). In this case, the density matrix ρ(t) is a (N+1)×(N+1) matrix with the following elements:(10)$$\rho_{0,0}=\langle 0|\rho |0\rangle \;,\;\rho_{0,\alpha }=\langle 0|\rho |\alpha \rangle =\rho_{\alpha ,0}^{*}\;\;,\;\;\rho_{\alpha ,\beta }=\langle \alpha |\rho |\beta \rangle =\rho_{\beta ,\alpha }^{*}$$
(α,β=1,2,…,N). The explicit evolution equations of the density matrix elements, as derived from the master Equation (Equation 7), are given in Appendix A. A special yet broad subclass of initial states relevant to analytically describe the non-normal ME is provided by the mixed states:(11)$$\rho_{i}=\rho \left(t=0\right)=p|\psi_{0}\rangle \langle \psi_{0}|+\left(1-p\right)|0\rangle \langle 0|\;,$$
where $|\psi_{0}\rangle =\sum_{\alpha =1}^{N}c_{\alpha }^{\left(0\right)}|\alpha \rangle$, $\sum_{\alpha =1}^{N}{\left|c_{\alpha }^{\left(0\right)}\right|}^{2}=1$ and 0≤p≤1. For this subclass of initial states of the system, the evolved density operator ρ(t), as obtained from Equations (A3)–(A5) given in Appendix A, can be written in the following form:(12)$$\rho \left(t\right)=p|\psi \left(t\right)\rangle \langle \psi \left(t\right)|+\rho_{0,0}\left(t\right)|0\rangle \langle 0|.$$
In the above equation, we have set(13)$$|\psi \left(t\right)\rangle =\exp\left(-iH_{eff}t\right)|\psi_{0}\rangle =\sum_{\alpha =1}^{N}c_{\alpha }\left(t\right)|\alpha \rangle \;,\;\;\;\rho_{0,0}\left(t\right)=1-p\sum_{\alpha }{\left|c_{\alpha }\left(t\right)\right|}^{2}$$
with $c_{\alpha }\left(t=0\right)=c_{\alpha }^{\left(0\right)}$. The state |ψ(t)〉 evolves solely under the effect of the effective non-Hermitian Hamiltonian (quantum jumps do not affect its evolution), and the amplitudes $c_{\alpha }\left(t\right)$ are thus found by solving the following linear system of coupled equations:(14)$$\begin{array}{ccc} i\frac{dc_{\alpha }}{dt} & = & \sum_{\beta =1}^{N}\langle \alpha |H_{eff}|\beta \rangle c_{\beta }\left(t\right)=\left(\omega_{\alpha }-\omega_{0}-i\frac{\gamma_{L}+\gamma_{R}}{2}\right)c_{\alpha }+\\ & - & i\gamma_{L}\sum_{\beta >\alpha }\exp\left[ik\left(x_{\beta }-x_{\alpha }\right)\right]c_{\beta }-i\gamma_{R}\sum_{\beta <\alpha }\exp\left[ik\left(x_{\alpha }-x_{\beta }\right)\right]c_{\beta }. \end{array}$$
The distance from equilibrium d(t), as measured by the Frobenius distance $d\left(t\right)=d_{F}\left(t\right)$, can be readily calculated from Equation (Equation 12) and reads as follows:(15)$$d\left(t\right)=\sqrt{2}pP_{s}\left(t\right)\;,$$
where we has set(16)$$P_{s}\left(t\right)=\sum_{\alpha =1}^{N}{\left|c_{\alpha }\left(t\right)\right|}^{2}.$$
The quantity $P_{s}\left(t\right)$ can be viewed as the survival probability of atomic excitation, i.e., the probability that no photon has been irreversibly emitted into the waveguide at time *t* [51]. Since we assume that there are not atom–photon bound states, clearly $P_{s}\left(t\right)\to 0$ as t→∞, corresponding to d(t)→0 as t→∞. However, the decay behavior of $P_{s}\left(t\right)$, and thus of d(t), can display anomalous features related to the non-normal nature of the effective non-Hermitian Hamiltonian $H_{eff}$ (see, e.g., [51,52]). Departure from non-normality of the Hamiltonian $H_{eff}$ can be measured by the Henrici parameter [50,72]:(17)$$\nu ={\left(\sum_{\alpha ,\beta =1}^{N}{\left|{\left(H_{eff}\right)}_{\alpha ,\beta }\right|}^{2}-\sum_{\alpha =1}^{N}{\left|\xi_{\alpha }\right|}^{2}\right)}^{1/2}\;,$$
where ${\left(H_{eff}\right)}_{\alpha ,\beta }=\langle \alpha |H_{eff}|\beta \rangle$ and $\xi_{\alpha }$ are the *N* eigenvalues of the matrix ${\left(H_{eff}\right)}_{\alpha ,\beta }$. The Henrici parameter ν vanishes if and only if the matrix ${\left(H_{eff}\right)}_{\alpha ,\beta }$ is normal. Effective non-Hermitian Hamiltonians in chiral waveguides, where $\gamma_{R}\neq \gamma_{L}$, typically exhibit a high degree of non-normality. However, some level of non-normality can also arise in non-chiral waveguides. As an example, Figure 2a shows the non-normality—quantified by the Henrici parameter ν—of the effective Hamiltonian $H_{eff}$ as a function of the ratio $\gamma_{R}/\gamma_{L}$ for a system of N=8 emitters with identical resonance frequencies $\omega_{n}=\omega_{0}$. The emitters are equally spaced along the waveguide with spacing $d=x_{n+1}-x_{n}$ chosen such that kd=2π. For this value of kd, no atom–photon bound states are present, meaning all eigenvalues $\xi_{\alpha }$ have strictly negative imaginary parts—except in the special case of a non-chiral waveguide ($\gamma_{L}=\gamma_{R}$), where bound states do emerge. Note that the largest value ν of non-normality is obtained in the chiral *cascaded* configuration [58] $\gamma_{R}/\gamma_{L}=0$, corresponding to unidirectional photon emission, while a normal matrix (ν=0) is found in the limit of a non-chiral waveguide $\gamma_{R}=\gamma_{L}$. However, for different values of the phase kd, non-normality is obtained for a non-chiral waveguide $\gamma_{L}=\gamma_{R}$ as well.

Non-normality is responsible for a variety of transient anomalous relaxation of $P_{s}\left(t\right)$ toward zero, such a *quiescent dynamics*, corresponding to an almost non-decaying probability $P_{s}\left(t\right)≃1$ for a long time interval 0<t<τ, after which the decay starts with an abrupt drop of the survival probability [52]. For a given initial condition $\left\{c_{\alpha }^{\left(0\right)}\right\}$, the upper bound for $P_{s}\left(t\right)$ is given by $P_{s}\left(t\right)\leq \sigma_{m}\left(t\right)$, where $\sigma_{m}\left(t\right)$ is the largest eigenvalue of the matrix $A^{†}\left(t\right)A\left(t\right)$ with $A\left(t\right)=\exp\left(-iH_{eff}t\right)$ [51]. For a fixed time t>0, the largest value $\sigma_{m}\left(t\right)$ is assumed for the initial condition $\left\{c_{\alpha }^{\left(0\right)}\right\}$ of the system, which is the eigenvector of $A^{†}\left(t\right)A\left(t\right)$ with eigenvalue $\sigma_{m}\left(t\right)$. The resilience time τ can be defined as the largest time t=τ such that the upper bound $\sigma_{m}$ of the survival probability remains larger than a reference value $P_{b}∼1$ in the entire time interval (0,τ). An example of resilience dynamics in a chiral waveguide, with N=8 emitters, phase kd=2π, and chiral parameter $\gamma_{R}/\gamma_{L}=0.1$, is shown in Figure 2b. On the other hand, for other initial states, such as when $c_{\alpha }\left(0\right)=\delta_{\alpha ,1}$ (this condition corresponds to excitation of the atom at the far left edge of the chain), resilience dynamics are not observed, and the decay is almost exponential (as $\gamma_{R}/\gamma_{L}\to 0$) with a decay rate $\gamma_{L}$. These distinct decay behaviors depend on the initial state preparation and can yield a pronounced non-normal ME, as illustrated in Figure 3. The figure depicts the numerically computed behaviors of the distance d(t), either the trace distance $d\left(t\right)=d_{T}\left(t\right)$ (Figure 3a) or Frobenius distance $d=d_{F}\left(t\right)$ (Figure 3b), corresponding to three different initial states (Equation 11) in a highly chiral waveguide ($\gamma_{L}=1$, $\gamma_{R}=0.1$). State I is defined by Equation (Equation 11) with p=1 and $c_{\alpha }=\delta_{\alpha ,1}$. This is a pure state with the atom at the far left end of the chain (α=1) in the excited state and all other atoms in the ground state. This state displays a near exponential relaxation to equilibrium with a relaxation rate $≃\gamma_{L}$; the weak oscillations observed in the decay on a log scale, shown in the insets, arise from weak interference effects originating from a non-vanishing yet small value $\gamma_{R}$.

State II is defined by Equation (Equation 11) with p=0.5 and with $c_{\alpha }$ given by the distribution shown in the lower inset of Figure 2b. This is a mixed state where with 50% probability, the system is already in the equilibrium state (all atoms are in the ground state), and with 50% probability, the atoms are prepared in the single-excitation entangled state shown in Figure 2b. This state displays resilience to decay for a time interval τ∼16, leading to a pronounced ME when compared to the relaxation of state I. Finally, state III is defined by Equation (Equation 11) with p=0.5 and $c_{\alpha }=\delta_{\alpha ,8}$. This is a mixed state where with 50% probability, the system is already in the equilibrium state (all atoms are in the ground state), and with 50% probability, the atom at the far right end of the chain (α=N=8) is in the excited state. Compared to state II, the initial preparation of atoms in state III is clearly much more feasible from an experimental standpoint. Non-normal ME is also observed when comparing the relaxation dynamics of states I and III. Finally, we note that in the limiting case of cascaded non-chiral waveguides, where $\gamma_{R}/\gamma_{I}=0$ [58], the effective Hamiltonian $H_{eff}$ becomes non-diagonalizable and exhibits a higher-order exceptional point when $\omega_{n}=\omega_{0}$. In this regime, the expansion analysis must include generalized eigenvectors due to the defective nature of both L and $H_{eff}$ (see, e.g., [51,52]). Nevertheless, non-normality—the fundamental mechanism underlying the non-normal memory effect—remains present. Numerical simulations show a relaxation behavior qualitatively similar to that in Figure 3, except that the weak oscillations in the decay dynamics, stemming from interference between nearby resonances, are completely suppressed. In other words, exceptional points are not necessary for the observation of the non-normal ME; the essential ingredient is non-normality itself.

## 4. Conclusions

In this work, we have identified and characterized a novel mechanism underlying the quantum Mpemba effect, arising from the non-normality of the Liouvillian superoperator in Markovian open quantum systems. Unlike previously known mechanisms—such as strong Mpemba effects tied to spectral suppression, or those relying on non-Markovian memory or initial correlations—our results demonstrate that transient interference between non-orthogonal decay modes can alone induce anomalous relaxation behavior. In particular, we have shown that certain initial states, despite being closer to thermal equilibrium, may exhibit slower early-time relaxation than states prepared farther away, purely due to non-normal dynamics. We illustrated this phenomenon using a waveguide QED model, a natural platform where directionality and radiative coupling introduce non-normality in a controlled manner. Waveguide QED setups involving quantum dots, trapped atoms, and superconducting artificial atoms coupled to one-dimensional photonic structures have already been realized and studied in various contexts [64,65,66,67,68,69,70,71]. These platforms provide the necessary ingredients—namely, directional photon emission and tunable emitter spacing—to explore non-normal dynamics and transient relaxation effects, as discussed in this work. Our analysis reveals clear signatures of the non-normal quantum Mpemba effect, such as transient freezing and delayed relaxation, even in the absence of memory effects or asymptotic spectral degeneracies. These results broaden the theoretical understanding of quantum nonequilibrium phenomena and point to non-normality as a universal and designable resource for controlling thermalization dynamics. While our analysis focuses on quantum systems, we note that the non-normality mechanism underlying the Mpemba effect is also relevant in classical settings [47,48,49,50], where the transition matrix of the master equation is generally non-normal. This suggests that similar transient behavior may arise in classical stochastic processes as well [50].

Looking forward, this work provides several promising future directions. On the theoretical side, it suggests a need to systematically classify non-normal contributions to quantum relaxation and to develop tools for diagnosing them in complex systems. Experimentally, waveguide QED platforms—both in photonic and superconducting circuit architectures—offer feasible testbeds to probe and exploit non-normal Mpemba behavior. Ultimately, harnessing such effects may provide new knobs for quantum state preparation, information processing, and thermodynamic control in engineered quantum environments.

## Figures and Tables

**Figure 1 entropy-27-00581-f001:**
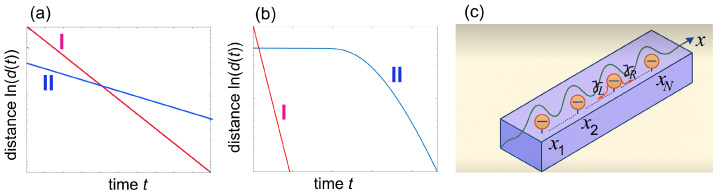
(**a**,**b**) Typical behaviors of the distance *d* (in log scale) versus time *t* (**a**) in the strong Mpemba effect and (**b**) in the non-normal Mpemba effect. In the strong ME, state I is initially further from equilibrium than state II; however, it displays a larger decay rate. In the non-normal ME, the asymptotic long-time decay rates of states I and II are the same; however, state II displays anomalously slow decay at initial times, resulting in a pronounced ME. (**c**) Schematic of a set of *N* point-like quantum emitters (two level atoms) embedded in a one-dimensional photonic waveguide. The spontaneous emission decay rates of the single quantum emitter in the left and right waveguide propagating modes are $\gamma_{L}$ and $\gamma_{R}$, respectively. A chiral waveguide corresponds to $\gamma_{L}\neq \gamma_{R}$.

**Figure 2 entropy-27-00581-f002:**
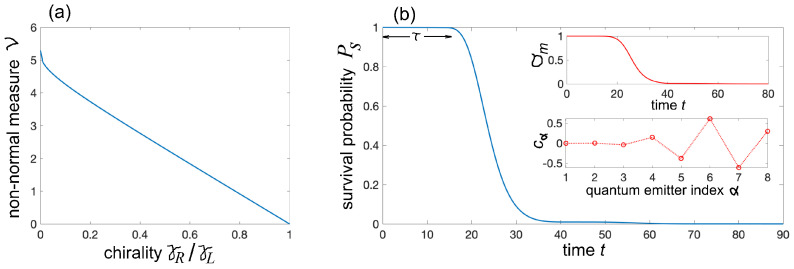
(**a**) Behavior of non-normality of the effective non-Hermitian matrix ${\left(H_{eff}\right)}_{\alpha ,\beta }$ (Henrici parameter ν) versus the ratio $\gamma_{R}/\gamma_{L}$ (chirality) in the waveguide QED setup of Figure 1c comprising N=8 equally spaced point-like quantum emitters. Parameter values are $\gamma_{L}=1$, kd=2π and $\omega_{n}=\omega_{0}$. (**b**) Numerically computed behavior of the survival probability $P_{s}\left(t\right)=\sum_{\alpha =1}^{N}{\left|c_{\alpha }\left(t\right)\right|}^{2}$ versus time *t* for the initial state $|\psi_{0}\rangle =\sum_{\alpha =1}^{N}c_{\alpha }\left(0\right)|\alpha \rangle$ with amplitudes $c_{\alpha }\left(0\right)$ depicted in the lower inset of panel (**b**). Parameter values are N=8, kd=2π, $\omega_{n}=\omega_{0}$, $\gamma_{L}=1$, and $\gamma_{R}/\gamma_{L}=0.1$. Note that the survival probability displays anomalous relaxation dynamics, with a nearly suppressed decay for a resilience time interval τ∼16, followed by a rapid drop toward zero. The upper inset in (**b**) shows the numerically computed behavior of $\sigma_{m}\left(t\right)$ versus *t*, where $\sigma_{m}\left(t\right)$ is the largest eigenvalue of $A^{†}\left(t\right)A\left(t\right)$ and $A\left(t\right)=\exp\left(-iH_{eff}t\right)$.

**Figure 3 entropy-27-00581-f003:**
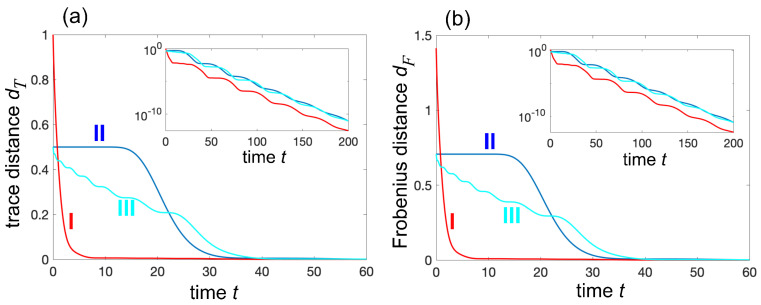
Illustrative examples of non-normal Mpemba effect in a waveguide QED setup. The plots show the numerically computed behavior of the relaxation dynamics, as measured by (**a**) trace distance and (**b**) Frobenius distance, for three different initial states $\rho_{i}$ and for the same parameter values as in Figure 2b. The insets in the two panels depict the decay behaviors of the trace and Frobenius distances on a vertical log scale. State I is defined by Equation (Equation 11) with p=1 and $c_{\alpha }=\delta_{\alpha ,1}$. This is a pure state with the atom at the far left end of the chain (α=1) in the excited state and all other atoms in the ground state. State II is a mixed state, given by Equation (Equation 11) with p=0.5 and with $c_{\alpha }$ given by the distribution shown in the lower inset of Figure 2b. This distribution is obtained as the eigenvector of $A{\left(t\right)}^{†}A\left(t\right)$ with t=16. State II displays resilience to decay for a time interval τ∼16, leading to a pronounced ME when compared to the relaxation of state I. State III is defined by Equation (Equation 11) with p=0.5 and $c_{\alpha }=\delta_{\alpha ,8}$. This is a mixed state where with 50% probability, the system is already in the equilibrium state (all atoms are in the ground state), and with 50% probability, the atom at the far right end of the chain (α=N=8) is in the excited state. Non-normal ME is also observed when comparing the relaxation dynamics of states I and III.

## Data Availability

No data were generated or analyzed in the presented research.

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
