# Peer review of "Quantum Mpemba Effect from Non-Normal Dynamics"

_entropy, 2025, doi:10.3390/e27060581_

Round 1

Reviewer 1 Report

Comments and Suggestions for Authors

In the manuscript, the author proposes a new mechanism of the quantum Mpemba effect, termed as the non-normal Mpemba effect.

Introduction is clearly written, and a brief pedagogical review of the quantum Mpemba effect is given in Section 2. In Section 3, the non-normal Mpemba effect is demonstrated in a waveguide QED system, which is experimentally relevant. It is shown that the predicted non-normal Mpemba effect actually occurs in such systems.

I think that this work reports interesting new results, and am happy to recommend the publication of this manuscript.

Although this work focuses on quantum systems, I suggest to include brief discussion that the mechanism of the non-normal Mpemba effect is also relevant in the setup of classical stochastic processes. In general, the transition matrix in the classical master equation is not normal, and   hence this mechanism is not limited to quantum systems.

typo:  In line 194, \gamma_L and \gamma_R are wrongly presented.

Reviewer 2 Report

Comments and Suggestions for Authors

In this paper, the author unveils a distinct mechanism rooted in the non-normal nature of the Liouvillian superoperator in Markovian dynamics. When the Liouvillian’s eigenmodes are non-orthogonal, transient interference between decaying modes can induce anomalous early-time behavior– such as delayed thermalization or transient freezing– even though asymptotic decay rates remain unchanged. This differs fundamentally from strong Mpemba effects, which hinge on suppressed overlap with slow-decaying modes. The author demonstrates this mechanism using a waveguide quantum electrodynamics model, where quantum emitters interact with the photonic modes of a one-dimensional waveguide. The directional and radiative nature of these couplings naturally introduces non-normality into the system’s dynamics. As a result, certain initial states– despite being closer to equilibrium– can exhibit slower relaxation at short times. This work reveals a previously unexplored and universal source of Mpemba-like behavior in memoryless quantum systems, expanding the theoretical framework for anomalous relaxation and opening new avenues for control in engineered quantum platforms.

These results make sense and sound interesting, and I believe can be of interest to entropy readers as well. I can therefore recommend this manuscript for publication in entropy once the authors address the following points:

(1)There are some punctuation errors. For example, there is no punctuation after formulas (4) ,(7) and (8), etc. I suggest the authors examine the manuscript carefully.

(2)In the first paragraph of page 1, the authors cite 34 references in line 27. I think if the authors write more details here and try to disperse some of the references, the readers will be clearer.

(3)There are some grammatical errors, such as “they” in line 79 of page 2 should be changed to “it”, and so on. I recommend the authors to correct these errors to make the paper more clear for readers to read.

(4)The authors are encouraged to briefly address how this scheme can be implemented in current experimental setups and provide relevant references to support the discussion.

Round 2

Reviewer 1 Report

Comments and Suggestions for Authors

The author addressed my comment and the manuscript is improved. I think it's ready for publication in Entropy.

Reviewer 2 Report

Comments and Suggestions for Authors

The authors have done a good job in responding to my questions. I therefore recommend publication.